# Silver Nanoparticles In Situ Synthesized and Incorporated in Uniaxial and Core–Shell Electrospun Nanofibers to Inhibit Coronavirus

**DOI:** 10.3390/pharmaceutics16020268

**Published:** 2024-02-14

**Authors:** Camila F. de Freitas, Paulo R. Souza, Gislaine S. Jacinto, Thais L. Braga, Yara S. Ricken, Gredson K. Souza, Wilker Caetano, Eduardo Radovanovic, Clarice W. Arns, Mahendra Rai, Edvani C. Muniz

**Affiliations:** 1Department of Chemistry, Federal University of Santa Catarina, Florianópolis 88040-900, Brazil; 2Department of Chemistry, State University of Maringá, Av. Colombo, 5790, Maringá 87020-900, Brazil; 3Laboratory of Virology, Institute of Biology, University of Campinas–UNICAMP, Campinas 13083-970, Brazil; 4Chemistry Institute, State University of Campinas, UNICAMP, Rua Josué de Castro Cidade Universitária, Campinas 13083-970, Brazil; 5Department of Microbiology, Nicolaus Copernicus University, 87-100 Torun, Poland; 6Department of Chemistry, Federal University of Piauí, Campus Ministro Petronio Portella, Ininga, Teresina 64049-550, Brazil; 7Department of Chemistry, Federal University of Technology-Paraná, Estrada dos Pioneiros, 3131, Londrina 86036-370, Brazil

**Keywords:** electrospun, coaxial nanofiber, metal nanoparticles, nanofiber matrices, mouse hepatitis virus (MHV-3)

## Abstract

In the present study, we sought to develop materials applicable to personal and collective protection equipment to mitigate SARS-CoV-2. For this purpose, AgNPs were synthesized and stabilized into electrospinning nanofiber matrices (NMs) consisting of poly(vinyl alcohol) (PVA), chitosan (CHT), and poly-ε-caprolactone (PCL). Uniaxial nanofibers of PVA and PVA/CHT were developed, as well as coaxial nanofibers of PCL[PVA/CHT], in which the PCL works as a shell and the blend as a core. A crucial aspect of the present study is the in situ synthesis of AgNPs using PVA as a reducing and stabilizing agent. This process presents few steps, no additional toxic reducing agents, and avoids the postloading of drugs or the posttreatment of NM use. In general, the in situ synthesized AgNPs had an average size of 11.6 nm, and the incorporated nanofibers had a diameter in the range of 300 nm, with high uniformity and low polydispersity. The NM’s spectroscopic, thermal, and mechanical properties were appropriate for the intended application. Uniaxial (PVA/AgNPs and PVA/CHT/AgNPs) and coaxial (PCL[PVA/CHT/AgNPs]) NMs presented virucidal activity (log’s reduction ≥ 5) against mouse hepatitis virus (MHV-3) genus Betacoronavirus strains. In addition to that, the NMs did not present cytotoxicity against fibroblast cells (L929 ATCC^®^ CCL-1TM lineage).

## 1. Introduction

The pandemic caused by the coronavirus (SARS-CoV-2) has become one of the main problems of public health worldwide in recent years. The alarming data so far point to approximately 750 million cases and more than 6.8 million deaths [1]. The highly contagious virus has a fever, coughing, sneezing, and sore throat as its main symptoms. However, severe conditions can lead to serious complications such as pneumonia, shortness of breath, and acute respiratory problems [2].

Because of this, in the last three years, the scientific community has been firmly committed to developing strategies capable of mitigating SARS-CoV-2. Among the approaches, the development of therapeutic agents; vaccines; rapid, sensitive, and specific diagnostic tools; preventive measures; and sanitizers stand out [3]. In this scenario, nanotechnology has been used as a powerful shield against the coronavirus [4]. Nanomaterials generally present sizes from 1 to 100 nm and have a high surface-to-volume ratio [5]. Thanks to their small size, they achieve unique physical–chemical and electrical properties that have been exploited in various fields of application, including biomedical fields [6]. Thus, they can act in the biotransport and bioavailability of antiviral drugs, thus reducing side effects, being able to codeliver several drugs, increasing their stability, and assisting the sustained release and specific delivery of antivirals [7]. They can also act in the development of vaccines and in acquiring highly specific and selective sensors [8]. In addition to applications in drug delivery systems, nanoparticles have also been used in prevention and disinfection strategies [9]. In this field, the development of personal protective equipment (PPE) and collective protective equipment (CPE), such as masks, lab coats, curtains, gloves, hospital sheets, and others, has attracted much attention [10]. Moreover, some nanoparticles have intrinsic antimicrobial characteristics, such as metallic silver nanoparticles (AgNPs), whose impact has been observed in the fight against SARS-CoV-2 [11].

In the biomedical field, a determining factor of the antimicrobial activity of AgNPs is the surface area/volume ratio. Thus, the interference of small-size nanoparticles on viral replication within the host cell is an advantage over other non-nanomaterials [12]. Regarding antiviral activity, AgNPs have unique virus-targeting characteristics and can easily bind to viral surface glycoproteins [13]. Then, AgNPs can enter the virus cells to exhibit virucidal activity by attacking the viral genome [14]. Given the antiviral and virucidal potential of AgNPs, some studies have investigated their inhibition application in PPE and CPE [15]. In this regard, the synthesis and incorporation of AgNPs in nanofiber matrices (NMs) obtained by electrospinning were investigated in the present study.

Nanofibers are polymeric filaments with diameters smaller than one μm [16]. These ultrathin materials have unique characteristics, such as high surface area, porosity, and flexibility [17]. Thus, they are promising for applications in the biomedical field, tissue engineering, industrial processes, media high-efficiency filters, protective clothing, catalysts, adsorbent materials, sensors, and energy storage [18]. Furthermore, incorporating AgNPs into nanofiber matrices (NMs) expands their potential applications in this field. This includes enhancing the performance of face masks for improved viral protection, reinforcing the durability and barrier properties of protective clothing, augmenting the efficiency of air filters for virus removal, promoting antimicrobial functionalities in medical dressings, contributing to the development of antiviral surfaces, and supporting enhanced water filtration systems. These advancements address a broad spectrum of preventive and protective measures in various settings.

In this sense, several techniques for manufacturing nanofibers have been developed, such as air jet spinning, centrifugal spinning, phase separation, electrospinning, and others [19]. Among them, electrospinning stands out due to its simple production scheme, reproducibility, and control over the diameter and morphology of the nanofibers [20]. In general, the electrospinning system consists of four essential components: (1) a high-voltage source, (2) an infusion pump, (3) a capillary tube (a small-diameter needle), and (4) a grounded collection plate, as shown in Figure 1A.

In a typical process, a high voltage is applied to cause a droplet of the solution to form at the tip of the needle. The electrified droplet is deformed, thus leading to the formation of a Taylor cone [19]. Once a Taylor cone is formed and exposed to a strong electric field, it becomes unstable, and a single fluid jet abruptly erupts, thereby accelerating toward the collector. As the jet travels through the air, the solvent evaporates, and the sample stretches beyond its original length, thus producing ultrathin, continuous polymeric nanofibers. Obtaining uniform nanofibers makes it necessary to optimize a series of parameters, including the choice of polymer, concentration, solvent, applied voltage, the flow rate of the solution from the syringe, and even the distance from the sample up to the collector [19].

In the present study, we sought to obtain uniaxial (Figure 1A) and coaxial polymeric nanofibers (Figure 1B) via electrospinning. For this, poly(vinyl alcohol) (PVA) and chitosan (CHT) were used to obtain uniaxial nanofibers and the core of coaxial nanofibers. Poly-ε-caprolactone (PCL) was used as a shell (coating) on the coaxial nanofibers. In addition, a differential of this study is related to the in situ synthesis and incorporation of the AgNPs, thereby avoiding the use of toxic reducing agents. In this scenario, the AgNPs were acquired in situ, thus obviating the need for postloading, as has been commonly observed in the majority of the literature [21]. Therefore, the PVA acts as a reducing and stabilizing agent for the Ag nanoparticles. Uniaxial and coaxial nanofibers obtained from biocompatible polymers can be applied both in the therapeutic field and in obtaining PPE and CPE to face SARS-CoV-2.

## 2. Material and Methods

### 2.1. Materials

Chitosan 85% deacetylated and presenting viscometric molar mass (Mv) of 87 kDa was purchased from Golden-Shell Biochemical (Taizhou, China). Poly(vinyl alcohol) (PVA) (Mw = 104.5 kDa) with a degree of hydrolysis 87–89% was purchased from Neon (Suzano, SP, Brazil). Poly-ε-caprolactone (PCL) with molar mass 80 kDa, silver nitrate (AgNO_3_), dimethylformamide (DMF), dichloromethane (DCM), and acetic acid used, and all were obtained from Sigma Aldrich. L929-NCTC clone 929 [L cell, L-929, derivative of Strain L] was acquired from ATCC^®^ (Manassas, VA, USA). DMEM (Eagle’s Minimal Essential Medium), FBS (fetal bovine serum), and Trypsin-EDTA were obtained from Gibco™ (Toronto, ON, Canada) and were used in the analysis of virucidal activity. All reagents were used without prior purification, and the water employed in all stages was ultrapure.

### 2.2. Methods

#### 2.2.1. Preparation of Polymeric Solutions for Electrospinning

All the experimental conditions to obtain the solutions of PCL, CHT, and PVA are shown in Table 1. It is noteworthy that other concentrations of PVA, CHT, and PCL were also investigated in the initial screening tests. However, only the optimized samples will be presented.

From the solutions presented in Table 1, PVA, PVA/AgNPs, and PCL nanofibers were obtained. The PVA/CHT polymeric blend was prepared in a 70:30 (*v/v*) ratio for solutions 3 and 2 (Table 1), respectively, presented in Table 1. For this blend formation, the mixture was kept under magnetic agitation for 1 h at 25 °C. The blend of PVA/CHT/AgNO_3_ was obtained similarly but employing solutions 4 and 2 from Table 1.

Coaxial nanofibers were prepared by using PCL (solution 1) as a shell and PVA/CHT or PVA/CHT/AgNO_3_ blends as a core to obtain PCL[PVA/CHT] or PCL[PVA/CHT/AgNPs], respectively.

#### 2.2.2. Uniaxial Electrospinning

To obtain the uniaxial nanofibers, each previously prepared solution was carefully transferred to a conventional 10 mL syringe of glass (internal diameter equal to 14 mm). In addition, the syringes were attached to a metallic capillary (needle) whose tip was previously cut, with an internal diameter equal to 0.07 mm. The mixture was kept in the syringe under constant pressure with a flow of 0.65 mL h^−1^. To obtain the NMs, a voltage of 23 kV was applied between the needle tip and the stainless-steel collector (diameter = 25 cm), separated by 10 cm. It is worth noting that AgNO_3_ was used as a precursor to obtain AgNPs, which were the principal antimicrobial agent in the formulation.

After obtaining the nanofibers containing “single polymers” (PVA, PVA/AgNPs, and PCL), the polymeric blends of PVA/CHT and PVA/CHT/AgNO_3_ were produced. The same electrospinning parameters were used to obtain the NMs of the blends. The matrices were carefully removed from the collector and stored in a desiccator under reduced pressure.

#### 2.2.3. Coaxial Electrospinning

In the coaxial nanofibers, the core was constituted by the polymeric blends of PVA/CHT and PVA/CHT/AgNO_3_, which were prepared as earlier informed in Section 2.2.1. The “shell” or coating comprised the PCL polymeric solution (Table 1). For coaxial electrospinning, it was necessary to use two syringes (Figure 1B)—one for the internal solution (core) and the other for the external solution (shell). Thus, for the combination of solutions, it was necessary to use a coaxial needle.

The coaxial needle employed in the present study was a “homemade” accessory designed by Professor Eduardo Radovanovic at the Department of Chemistry at the State University of Maringá and built by his staff. The accessory is presented in Appendix A.

The internal (core) and external (shell) solutions were transferred to two conventional 10 mL glass syringes. The mixture was kept in the syringes under constant pressure with a characteristic flow of 0.65 mL h^−1^. A voltage of 23 kV between the needle tip and the stainless-steel collector (diameter = 25 cm) was applied to obtain the fibers, which were separated by 10 cm. The matrices were carefully removed from the collector and stored in a desiccator under reduced pressure.

#### 2.2.4. Nanofiber Characterization

All produced uniaxial and coaxial nanofibers were examined by SEM images performed using a scanning electron microscope (Quanta 250, Tokyo, Japan) coupled with an energy-dispersive X-ray spectroscopy (EDS). The samples were sputter-coated with a thin layer of gold, thereby allowing for SEM visualization. Images were taken by applying an electron beam accelerating voltage of 10–20 kV. Infrared spectroscopies were performed in an FTIR spectrometer using an ATR accessory (Vertex 70v, Bruker, Billerica, MA, USA) operating from 4000 to 400 cm^−1^ and with a resolution of 4 cm^−1^. The spectrum was obtained after cumulating 128 scans for each acquisition. The crystalline structure of materials was investigated by using a wide-angle X-ray diffraction (WAXS) pattern (XRD-6000, Shimadzu, San Jose, CA, USA) equipped with Ni-filtered Cu-K (λ = 0.15418 nm), thus applying an accelerating voltage of 40 kV at 30 mA with 0.02° resolution and 2° min^−1^ scanning speed.

TGA/DTG analyses were carried out in a thermogravimetric analyzer (Netzsch, model STA 409 PG/4/G Luxx, Burlington, MA, USA) at a 10 °C min^−1^ rate under 20 mL min^−1^ N_2_ atmosphere and from 30 to 800 °C range.

#### 2.2.5. Mechanical Analysis

The mechanical properties of the NMs were evaluated in triplicate using a texturometer (Stable Micro Systems, TA-TX2, Godalming, UK) according to ASTM standard method D882-91 (1996) [22]. For this, specimens of 20 × 5 mm (l × w) were used, and the parameters measured were tensile strength (σb), Young’s modulus, and elongation at break (εb) at a fixed temperature of 25 ± 1 °C.

#### 2.2.6. Disintegration Assay

The disintegration assay of the NMs was carried out following the methodology proposed in the literature [23]. For this, nanofiber samples measuring 1 × 1 cm^2^ were placed in contact with 10 mL of a phosphate buffer saline of pH 7.4. The system temperature and agitation were set at 37 °C and 50 rpm, respectively. An aliquot of the sample was taken at specific time intervals. Each aliquot was analyzed using electronic absorption measuring the absorbance at 430 nm. The solutions were also evaluated using dynamic light scattering (Malvern Zetasizer Nano, Malvern, UK) to obtain the hydrodynamic diameter (D_H_), polydispersity index (PDI), and Zeta potential (ζ) of the released AgNPs.

#### 2.2.7. Virucidal Assay

The virucidal assays were carried out in an NB-2 laboratory (Biosafety Level 2) following the recommendations of ANVISA Art. 1 and Art. 3 of IN 04/13 and IN 12/16 and methodologies described in the standards BS ISO 21702:2019 [24] and BS 14476:2013+A2:2019 [25], with some adaptations. The assays were performed in four replicates (n = 4). Coronaviruses MHV-3 strain (MHV genus Betacoronavirus with same genus as species SARS-CoV-1, SARS-CoV-2, MERS, and others) titration was performed according to the Tissue Culture Infectious Doses 50% method (TCID50).

Assays were performed in quadruplicate (n = 4) using all the nanofibers. Previous titration of MHV-3 was performed according to the 50% Tissue Culture Infectious Doses (TCID50) method, in which 10 sequential base dilutions of the virus (10^1^ to 10^7^) were performed in sterile microplates of 96 wells. Subsequently, an aliquot of 100 μL of each dilution was transferred to a monolayer of cells of the L929-NCTC clone 929, which was previously prepared at a concentration equivalent to 3 × 10^5^ cells/mL. After 48 h of incubation at 37 °C with 5% CO_2_, the microplates were read in an inverted microscope in search of the cytopathic effect (CPE), and they were compared regarding the cellular and viral control. The aqueous dispersion of AgNPs was also evaluated and prepared according to the previous literature [26]; additional details can be found in the Appendix A. Before analyzing the virucidal activity, the maximum nontoxic concentration (CMNT) of the AgNPs’ suspension against L929 cells was determined, since the test substance must be active only against the virus and not against the cells.

For virucidal activity assays, the same sample of the AgNPs’ volume and virus suspension was a mixture left in microtubes for predetermined times (30’, 1 h, 8 h, 24 h, 48 h, and 64 h). To analyze whether there was an influence of environmental factors on the decline in the MHV-3 titer, since the inoculum remained thawed for up to 64 h, an aliquot of the viral suspension, without contact with the AgNP solution, was kept in microtubes for 24 h, 48 h, and 64 h and used as control. At the end of each of the interaction times between viruses and AgNPs, this mixture (100 µL test sample and 100 µL of virus previously titled) was added to a 96-well plate. The whole system was added to fibroblast cells, which were L929 cells (3 × 10^5^ cells/mL) previously grown in 96-well plates and incubated at 37 °C in an oven with 5% CO_2_ for 48 h. After incubation time, the plates were read using an inverted microscope in search of the characteristic CPE of the virus, and the titers were calculated based on the method of Reed and Muench, 1938 [27]. Finally, the results are expressed as a percentage of viral inactivation compared to the untreated viral control (virus titer).

The employed negative controls were the cell control (3 × 10^5^ cells/mL) in DMEM medium, without virus, and the test sample. The virus controls were the virus titration and cell culture in a DMEM medium. The positive test is related to the presence of the virus performed in each test sample and cell line in the DMEM medium.

## 3. Results and Discussion

As mentioned earlier, the initial screening of the present study involved the optimization of the polymeric concentrations that would be used. The concentrations evaluated for PVA, PCL, and CHT were generally determined based on the previous literature [28]. Figure 2 shows the SEM micrographs obtained for the optimized samples.

In general, the PCL fibers were obtained with a high degree of uniformity (Figure 2A) and an average diameter of 800 nm (Figure 2B), which is in agreement with the literature [29]. On the other hand, the nanofibers constituted solely by PVA (Figure 2C,D) presented lower average diameters, in the range of 200 nm, which are also compatible with the reported data [30]. However, as previously described, nanofibers constructed exclusively by CHT had several defects (beads) and high polydispersity in their diameters [31]. In this case, there was a strong influence on the molar mass, degree of deacetylation, purity, and origin of the polysaccharide [32]. In general, polymeric blends of CHT and poly(ethylene oxide) (PEO) were used to obtain NMs [33].

Therefore, the formation of NMs composed of PVA/CHT in different percentages was evaluated. Our results have shown that the best ratio was PVA/CHT 70/30 (*v*/*v*) or (74/26 *w*/*w*), which is in line with previously published data [28]. The PVA/CHT nanofibers (Figure 2E,F) were obtained with a degree of uniformity in terms of diameter, despite the presence of beads. In addition, the fibers presented very tiny diameters, in the order of 180 nm.

Given the set of results, both the PCL and the PVA/CHT nanofibers were obtained adequately using the coaxial electrospinning method. Furthermore, based on the diameter of the uniaxial nanofibers, it was concluded that PCL should be used in the coating, that is, in the “shell” of the coaxial nanofibers. On the other hand, the ultrafine PVA/CHT nanofibers were consistent with the necessary characteristics for the composition of the core of the coaxial nanofibers.

In this sense, the coaxial nanofibers were obtained using the “homemade” system shown in Appendix A. In general, using two solutions in coaxial electrospinning requires additional considerations regarding the viscosity, miscibility, conductivity, vapor pressures, and flow of the internal and external solutions [34]. In the case of a combination of PCL shell and PVA/CHT core, the selection of solvents for the polymers is an important and complex step that will determine the success or failure of the NM. It is worth noting that the low solubility between the internal and external solutions will enable the formation of nanofibers. The results obtained for the NMs containing a core made of PVA/CHT (70:30 *v*/*v*) and a corona constituted by PCL are shown in Figure 2G,H.

In general, the PCL [PVA/CHT] nanofibers took a certain degree of uniformity in terms of diameter (average diameter equal to 334 nm), despite the presence of beads. Furthermore, a certain degree of polydispersity was observed in the diameter of the nanofibers, in some cases reaching values four times greater than the average. However, this was the best condition obtained by adjusting the electrospinning parameters. In addition, the coaxial nanofibers had intermediate diameters between the uniaxial nanofibers, referring to the core and the corona. In general, the NMs presented appropriate properties for application in treating wounds, the reconstruction of the cutaneous extracellular matrix (ECM), the incorporation and release of drugs, or even in PPE/CPE production [16]. The photos of NMs obtained for PVA, PCL, PVA/CHT, and PCL [PVA/CHT] can be found in Appendix A.

Given the success in acquiring uniaxial and coaxial nanofibers, the step of synthesis and incorporation of the AgNPs were conducted. The nanofibers were achieved following the same parameters used in uniaxial electrospinning without AgNPs. In addition to that, silver nitrate was always added to the PVA solution (Table 1). The AgNP formation was confirmed by NMs with a yellowish color, characteristic of these nanoparticles [35], as shown in Figure 3.

As highlighted in the introduction, the incorporation of AgNPs is based on their antimicrobial potential reported in the literature. Undoubtedly, one of the main characteristics of the present study is the in situ synthesis of AgNPs, without toxic reducing agents or the need to perform posttreatment. In this case, silver nitrate acts as a silver precursor, while the polymer, PVA, acts in the reduction of silver ions and, at the same time, on the stabilization of the AgNPs, as schematized in Figure 3.

In its possible molecular configurations, PVA will have many-sided -OH groups, some even free from H-bonds. This “excess” of -OH groups facilitates the reduction of metal ions [36]. Firstly, the metal cation complexes [Ag^+^PVA] are formed in an intermediate stage with the polymer [36]. This complexation occurs due to nonbonding 2p^2^(O) hybrid electrons in such -OH groups, which are mainly responsible for the adsorption of metallic cations to form a complex [37].

An interesting fact observed experimentally is that although silver nitrate was added with PVA and kept above 80 °C under agitation (Table 1), the yellowish color, characteristic of AgNPs, only appeared in the NMs collected in the stainless-steel collector. Thus, the molecular stretching required to break the H-bonds will not be as effective in the solution. Therefore, it was assumed that only small silver clusters (nucleation points) would be formed in the PVA polymer solution. However, when the polymeric sample is injected into the electrospinning system, applying a high electric field will favor the necessary molecular stretching, which is consistent with the model shown in Figure 3. Thus, the breakup of the H-bonds will favor the occurrence of the Ag^+^→Ag reaction and the formation and growth of AgNPs. Under these conditions, the AgNPs grow in this short period, thereby highlighted by the yellow color in the electrospun NM. As presented below, the AgNPs formed in this methodology were considerably small (from 5 to 20 nm), which is a fact also justified by the Ag–PVA interface inhibiting the growth of massive particles and justifying the differential behavior found in the present study.

Thus, in this reaction model, metallic silver was formed from an intermediate product of a polymeric complex, as shown in Figure 3. The formation of AgNPs was favored by the molecular alignment caused by the high voltage applied during electrospinning. This formation was accompanied by a change in color to a yellowish tone; see Appendix A. Thus, the metallic ions were converted to Ag atoms and followed by coalescence as Ag clusters and by cluster growth to produce Ag particles.

In general, numerous studies can be found in the literature presenting the incorporation of AgNPs in nanofibers. Many even claim to use the in situ method [38,39]. However, when looking at AgNP incorporation methodologies, the most modern methods still employ postloading. In these methods, the nanoparticles are prepared independently from the nanofibers and then adsorbed to the NMs [40]. In traditional methods, nanofibers incorporating AgNPs are obtained using toxic reducing agents, such as tannic acid and sodium borohydride [41]. Another methodology widely used to obtain AgNPs in NMs involves the posttreatment of nanofibers, such as using heat with high temperatures, microwaves, or ultraviolet radiation [40,41]. Therefore, such traditional methods are more laborious and time-consuming, as they comprise many steps, reagents, and instrumentation, and they conflict with green chemistry principles.

Figure 4 shows the SEM images of uniaxial and coaxial nanofibers incorporated in situ with AgNPs.

The PVA/AgNPs nanofibers showed a high degree of uniformity regarding their diameter, absence of beads, and an average diameter of 398 nm. In addition, the yellow color of the fibers (Figure 4A–C) refers to the success in obtaining AgNPs in situ. In general, when comparing nanofibers obtained in the absence of Ag, an increase in fiber diameter is observed, and a reduction in polydispersity takes place.

Figure 4D,E show the SEM images for the produced PVA/CHT (70/30) nanofibers. CHT is a natural and biocompatible polymer extensively studied in ECM reconstruction, which also presents intrinsic antimicrobial effects [42]. The presence of the silver salt did not affect the production of the nanofibers, with homogeneous fibers having fewer defects than those observed in the absence of AgNO_3_ (Figure 2E). This salt presence can be related to the increase in the conductivity of the solution given the silver nitrate addition. Electrospinning requires a minimum electrical conductivity in solution to form nanofibers [43]. In general, the increase in the number of electrical charges causes an increase in the elongation capacity of the solution, thus favoring the formation of smooth fibers of a smaller diameter and more excellent uniformity [44].

Despite this, the diameter of the fibers practically doubled due to the presence of silver nitrate and the consequent formation of AgNPs (Figure 4F). This increase can be associated with the AgNPs inside the fibers, since it was impossible to observe them on their surface. It is worth noting that the yellow color indicates the formation of AgNPs in Appendix A.

The conductivity-enhancing effect was even more remarkable when we evaluated the PCL [PVA/CHT/AgNPs] coaxial nanofibers in Figure 4G,H concerning Figure 2G (without silver). In this case, there was a significant increase in uniformity and a reduction in fiber polydispersity. Again, the formation of AgNPs was also highlighted by obtaining nanofibers with a yellowish color (Appendix A). The average diameter of the nanofibers was slightly higher than the diameter of the nanofibers in the absence of silver nitrate (Figure 4I). Once more, the increase can be attributed to the presence of AgNPs, which in this case was not so significant because it is a coaxial nanofiber.

The AgNPs synthesized in the structure of the nanofibers were characterized by taking advantage of the fibers constituted by PVA/AgNPs being soluble in water. The AgNPs characterization is shown in Figure 5.

In general, AgNPs were quickly released from PVA nanofibers. Figure 5A shows the presence of an absorption peak with a maximum of 4 30 nm. In addition, a continuous increase in absorption intensity of up to 500 s was observed. The kinetic data, Figure 5B, shows a linear increase in the absorbance of AgNPs of up to 500 s. DLS further evaluated the release solution to determine the hydrodynamic diameter (D_H_), polydispersity index, and zeta potential (ζ) of the released AgNPs. The results obtained are shown in Figure 5C.

The released AgNPs had a D_H_ between 5 and 20 nm, an average D_H_ of 11.7 nm, and a polydispersity index of 0.4. According to the literature, this kind of nanoparticle presented ideal antimicrobial effects. Furthermore, as expected, the surface potential obtained was −5.1 ± 0.6 (mV), which was close to neutrality. Figure 5D shows the NM disintegration over time. Note the presence of a yellowish color, which is characteristic of spherical AgNPs.

The nanofibers obtained from the uniaxial and coaxial polymeric blends PVA/CHT/AgNPs and PCL [PVA/CHT/AgNPs], respectively, did not suffer disintegration. Thus, the presence of CHT and PCL in the nanofibers led to an increasingly significant hydrophobicity of the material, which is a positive fact for applications in PPE and CPE. It is noteworthy that the presence of chitosan in these nanofibers can contribute to the formation of nanoparticles, given its potential as a reducing and stabilizing agent for them [45].

The dissolution of the AgNPs and the release of Ag^+^ ions into the relevant medium represent a plausible mechanism for antimicrobial activity [46]. However, in terms of surface applications, the presence of immobilized nanoparticles capable of serving as reservoirs for silver ions is noteworthy. In this manner, immobilized AgNPs, distributed throughout ultrafine and porous material, continuously provide a sufficiently high concentration of silver antimicrobial species in their vicinity, thereby maintaining activity for several days, as is desired in PPE and CPE.

This effect can be highly advantageous in real-world application conditions, where fluid circulation on-site (from sneezing for example) leads to the release of active species. Thus, an immobilized nanoparticle near a microorganism can release several tens of thousands of silver atoms in its vicinity, thus creating a locally elevated concentration of antimicrobial agents (Trojan horse effect) [47], with information supported by in vitro assays.

The presence of silver in the nanofibers was also evaluated by EDS, as shown in Figure 6.

Figure 6A shows the presence of carbon, oxygen, and silver atoms, as well as the layer of gold that coated the sample. The EDS spectrum demonstrates the existence of Ag in the fiber (signal at 2.98 eV). PVA/CHT/AgNPs nanofibers, Figure 6B, in addition to carbon, oxygen, and silver, also presented nickel, copper, and zinc amounts, which were probably complexed with CHT [48]. In this case, the silver signal was even more significant, thus indicating a superior amount of Ag in the nanofibers. A similar fact was observed in Figure 6C for the PCL [PVA/CHT/AgNPs] coaxial nanofiber, with an intense signal at 2.98 eV.

The infrared spectra of PVA, CHT, PCL, PVA/AgNPs, PVA/CHT/AgNPs, and PCL [PVA/CHT/AgNPs] are shown in Figure 7.

In general, the main signals are highlighted in Figure 7A,B for the NMs and their precursors. The prominent characteristic peaks of PCL were observed at 2949 cm^−1^ (CH_2_ asymmetric stretch) and 1727 cm^−1^ (carbonyl stretch). The symmetric peaks of C-O-C were observed at 1170 cm^−1^, CH_2_ stretching at 2849 cm^−1^, C-O and C-C at 1293 cm^−1,^ and the asymmetric C-O-C at 1240 cm^−1^ [49]. In the case of PVA, the characteristics peaks observed were absorption peaks at 3337 cm^−1^ (O-H stretching), 2934 cm^−1^ (CH_2_ asymmetric stretching), 2911 cm^−1^ (CH_2_ symmetric stretching), 1727 cm^−1^ (C=O stretching), 1433 cm^−1^ (CH_2_ bending) and 1097 cm^−1^ (stretching of C-O and bending of OH) [50].

The signals referring to the CHT appeared at 895 and 1150 cm^−1^, thereby being associated with the vibration of the –COC– groups of the saccharides. The peaks at 1030 and 1080 cm^−1^ were attributed to the –CO stretching vibration, while those at 1652, 1587, and 1318 cm^−1^ were attributed to the –NH bending of the amine groups I, II, and III, respectively [51]. The symmetrical deformation mode of the –CH_3_ group appeared at 1374 cm^−1^, and the elongation mode for the –CH bond appeared at 2922 cm^−1^. Finally, the broad band at approximately 3430 cm^−1^ corresponds to the –OH stretching vibration of the CHT.

The characteristic signs of the employed polymers were generally found in the blends and the coaxial nanofibers. Some of the main signals are highlighted in Figure 7B. For example, the signals at 2911 cm^−1^, 1374 cm^−1^, and 1240 cm^−1^ refer to PVA, CHT, and PCL, respectively.

The X-ray diffractograms obtained are shown in Figure 7C,D for precursors and NMs, respectively. The CHT diffractogram reveals the presence of a peak at 11.8° and another at 20.2°, thus representing the intrinsic symmetric regions of the CHT. The XRD of the PVA presented a signal around 20° corresponding to the semicrystalline nature of the pure PVA [52]. Finally, the PCL homopolymer crystallized readily, and the diffraction pattern reveals the presence of significant crystallinity, with peaks at 21.9° and 24.2° corresponding to the (110) and (200) planes, respectively, of the orthorhombic crystal structure [53].

The uniaxial nanofibers PVA/CHT, PVA/AgNPs, and PVA/CHT/AgNPs depicted in Figure 7D showed an amplified signal between 15° and 30°. In addition, signals from the CHT crystalline regions appeared discretely. Thus, the interactions between the polymeric chains in the blend broke the crystalline arrangements in the single polymer. Furthermore, it appears that the presence/formation of AgNPs further reduced the intensity of the broadband, thereby indicating its interaction with the polymeric chains. On the other hand, the XRD of the coaxial nanofibers (PCL [PVA/CHT/AgNPs]) presents a broadband between 15° and 30°, therein referring to the polymers constituting the core. In addition, coaxial NMs also presented intense signals at 21.9° and 24.2°, which are attributed to the intrinsic crystallinity of PCL [53]. Thus, PCL crystallinity indicates the efficient formation of the coaxial nanofiber, thus denoting the absence of interactions between the core and corona chains, which is a desired fact [34].

The thermal stability of the nanofibers and their precursors were evaluated by TGA in Figure 8A,C, and their respective DTG curves are shown in Figure 8B,D. The first stage of weight loss in the range of 30 to 120 °C in all the TGA profiles was attributed to the loss of crystalline water [54]. Thus, the first region between 50 and 200 °C can be attributed to the loss of absorbed water molecules. The second region, between 200 and 340 °C, is related to water loss bound to the polymer matrix. The third region between 340 and 450 °C is associated with polymer decomposition and carbonization [55]. For the CHT, degradation occurred above 250 °C with a T_max_ at 280 °C. For the PCL, a degradation step can be observed in the range of 350 to 450 °C, which agrees with the literature [56]. The results obtained for the uniaxial nanofibers (Figure 8A,B) confirm the presence of both polymers in the NM. In the case of the coaxial nanofibers, the peaks referring to all the constituent polymers were also verified according to the used proportions. The results confirm the nanofiber composition and interaction.

The mechanical properties of the obtained nanofibers are shown in Table 2.

In general, it is observed that the inclusion of chitosan in PVA nanofibers leads to a decrease in their mechanical performance, which is in line with the literature [57]. However, incorporating AgNPs increases Young’s modulus and elongation at break in PVA and PVA/CHT nanofibers [58]. Similar results were verified for the coaxial nanofibers, thus indicating that incorporating AgNPs improves the interaction between the polymeric materials of the blend, possibly due to their interaction, reduction, allocation, and “link” with both. It is also noted that PCL improves all the properties related to the nanofiber in the absence of the polymer, which is associated with the PCL’s intrinsic mechanical characteristics [59]. Thus, comparatively, coaxial NM is superior to uniaxial PVA/CHT/AgNPs nanofibers.

## 4. Virucidal Activity

In general, AgNPs present an essential role in the control of microorganisms, including against SARS-CoV-2 [13,15]. Thus, all the NMs with and without AgNPs were evaluated against the MHV-3, expressed as a percentage of viral inactivation compared to the untreated viral control, as shown in Appendix A.

According to the data presented in Appendix A, only the materials with a reduction percentage between 99.99% to 99.9999% (Log’s reduction ≥ 4) showed virucidal activity, as described by the Microchem Laboratory [60].

Antiviral agents are substances that directly inhibit the viral replication cycle within host cells. They can also prevent the release of viruses from cells to the intercellular space [61]. On the other hand, virucidal agents are substances that directly inactivate viral infectivity outside host cells. In the case of nanoparticles, antiviral mechanisms should target virus attachment, penetration, replication, and budding. These mechanisms are related to preventing virus binding to host cells and blocking viral replication, depending on the form and type of nanoparticles used. In general, nanoparticles can block the steps, thus changing the structure of the capsid protein. In this way, they can reduce virulence, which can be attributed to physical and chemical means of reducing the active viral load [62].

These compounds are capable of completely preventing the infectivity of cell-free viruses. In this context, NMs presenting virucidal activity seem to be very promising for the development of PPE and CPE. In this sense, the NM performances against the coronavirus (MHV-3) were evaluated and presented in Table 3.

First, it is worth noting that the evaluated NMs with or without AgNPs did not show cytotoxicity through the tests carried out with the L929 ATCC^®^ CCL-1TM lineage. However, the AgNP dispersion was toxic up to 1:1 dilution (low toxicity). For this reason, the virucidal tests for this sample were performed with an initial concentration of 1:10 AgNPs. This result is very promising, thus indicating that the incorporation of AgNPs in the NM corroborates with the improvement of their cytocompatibility. Concerning the antiviral activity, the nanofibers in the absence of AgNPs did not present virucidal activity in any case (log reduction < 4). However, comparing the nanofibers made of PVA with those containing CHT (uniaxial or coaxial), it is verified that CHT increases by 1 log on virus reduction. This increase may be associated with CHT, whose antimicrobial potential has been reported in the literature [63]. Nevertheless, the concentration range was insufficient to obtain a virucidal character sample. This increase depends on the contact time of the nanofibers with the culture medium. Thus, for uniaxial PVA/CHT nanofibers, a log reduction equal to 2 was obtained after 8 h of contact.

On the other hand, for PCL [PVA/CHT] coaxial nanofibers, a log reduction equal to 2 was only observed after 24 h of contact. This reduction may be associated with the external PCL layer in the coaxial nanofibers that will protect the core content (PVA/CHT). In this case, a longer interaction time with the aqueous medium is required, which will cause the material to swell and lead to viral inhibition.

Concerning AgNPs, we first evaluated the virucidal activity of the nanoparticles obtained in an aqueous medium following the method proposed by Turkevich [26]. The results show that the AgNPs have high virucidal potential since, with 30 min of incubation, a log reduction greater than 5 had already been observed. It is also noted that with the increase in time, there was a further increase, reaching 6.0. The virucidal activity of the AgNPs has been the target of several scientific works associated with SARS-CoV-2 [15,64].

The control test to verify the influence of environmental conditions on the MHV-3 titer did not point to a significant drop in the virus titer in 24 h and 48 h at room temperature, whereas in 64 h there was an average reduction of just over 1 log10. Even so, these findings confirm the role of AgNPs in the inactivation of viral particles.

To date, the viral inhibition mechanism of AgNPs on SARS-CoV-2 remains under study; however, important observations have already been obtained. It is known, for example, that AgNPs can interact with viral nucleic acids, thereby having intracellular antiviral action. Still, most studies highlight the ability of AgNPs to interact with structural proteins on the surface of extracellular viruses. This interaction is sufficient to damage surface proteins to affect their structural integrity [65]. Thus, AgNPs end up interfering with viral entry, being able to cleave disulfide bonds and binding to viral surface proteins rich in sulfhydryl groups to destabilize the protein, thus affecting viral infectivity [66]. In the specific case of SARS-CoV-2, AgNPs will lead to the cleavage of disulfide bonds in angiotensin-converting enzyme-2 (ACE2) receptors and the spike protein [13,67].

A crucial aspect of the effectiveness of the antiviral action is the metal nanoparticle. Thus, AgNPs with a diameter between 2 and 15 nm will have an antiviral effect, and those with a size around 10 nm in diameter appeared to be the most effective [68].

As seen in Figure 5, the AgNPs obtained in situ on the nanofibers showed sizes between 5 and 20 nm, within the range of most significant viral activity. The data in Table 3 show that all uniaxial and coaxial nanofibers incorporated with AgNPs proved to have virucidal activity even in the shortest time evaluated. Only a slight difference was observed between the NM, which again indicated that the presence of PCL coating the PVA/CHT/AgNPs hinders, to some extent, the contact of AgNPs with the virus. However, the mechanical and structural advantages associated with the presence of PCL justify its use. In this case, the choice of uniaxial or coaxial nanofiber will depend on the intended application.

The activity of silver in controlling the coronavirus has been evaluated in the literature [69]. In this scenario, the results presented in Table 3 confirm the potentiality of colloidal silver in controlling the virus. However, our work also verified the potentiality of nanofibers made up of uniaxial PVA/CHT and PCL [PVA/CHT-AgNPs] in SARS-CoV-2 treatment, prevention (PPE and/or CPE), and even for use in sanitizers. 

## 5. Conclusions

The results pointed to the effectiveness of the uniaxial and coaxial electrospinning matrices incorporated with AgNPs in developing materials designed for the treatment, prevention, and control of SARS-CoV-2. In general, the uniaxial and coaxial nanofibers were obtained with uniformity and diameters consistent with those needed for the desired type of application. AgNPs were synthesized in situ through an innovative approach that avoids the application of toxic reducing agents, postloading, or posttreatment of the obtained matrices. The AgNPs synthesized in situ by the action of PVA as both the reducing and stabilizing agent had an average size of 11.7 nm and a very low polydispersity, which is ideal for antiviral action. Characterizations using FTIR, DRX, SEM-EDS, and thermal analysis confirmed the structural polymeric composition, thus allowing for a further understanding of the interaction of the polymers. The nanofibers presented mechanical properties consistent with those desired for preventive and sanitizer agent applications, with consistent elasticity and high elongation at break. They stand out for providing comfort, adaptability, and efficacy during use. The developed NMs, with mechanical properties conducive to comfort and efficacy, hold promise for applications such as enhanced face masks, protective clothing, air filters, antimicrobial dressings, antiviral surfaces, and water filtration systems, thus showcasing their versatility in diverse domains. Finally, viral assays showed that nanofibers without AgNPs do not present virucidal activity, even those containing CHT. However, the presence of AgNPs led to significant virucidal effects even in the shortest incubation time (30 min). Thus, samples involving the in situ incorporation of AgNPs into uniaxial PVA/CHT and coaxial PCL [PVA/CHT] nanofibers have a high potential for application in mitigation measures for COVID-19 and other possible viral contaminants.

## Figures and Tables

**Figure 1 pharmaceutics-16-00268-f001:**
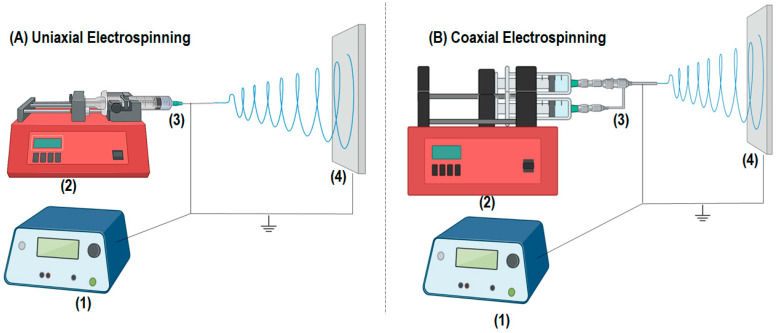
Representative scheme of a (**A**) uniaxial and (**B**) coaxial electrospinning system, where (1) is the high voltage source, (2) is the polymeric solution injection pump, (3) is the uniaxial or coaxial metallic needle, and (4) the grounded collection plate.

**Figure 2 pharmaceutics-16-00268-f002:**
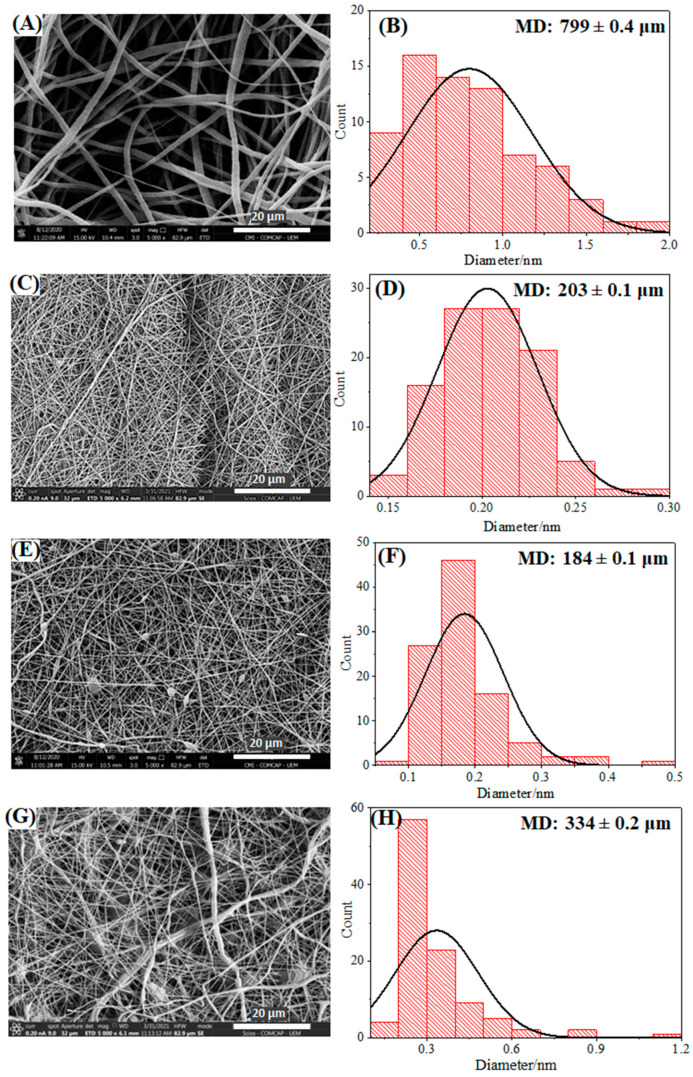
SEM image and frequency diameter distribution of nanofibers: (**A**,**B**) PCL, (**C**,**D**) PVA, (**E**,**F**) PVA/CHT 70:30, and (**G**,**H**) PCL [PVA/CHT] coaxial.

**Figure 3 pharmaceutics-16-00268-f003:**
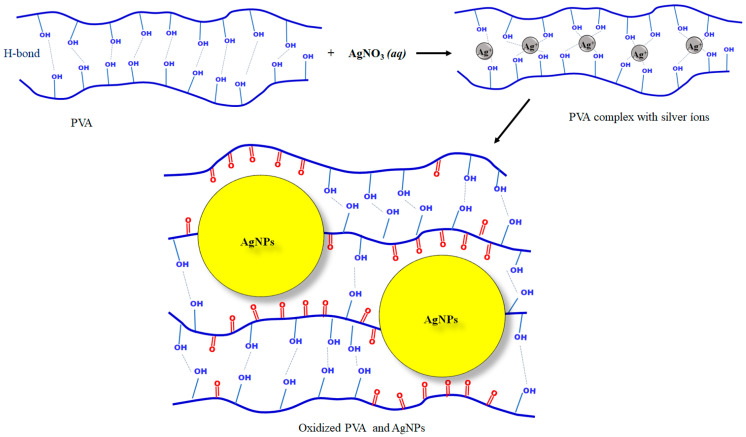
Representative scheme of forming AgNPs from the precursor silver nitrate and PVA.

**Figure 4 pharmaceutics-16-00268-f004:**
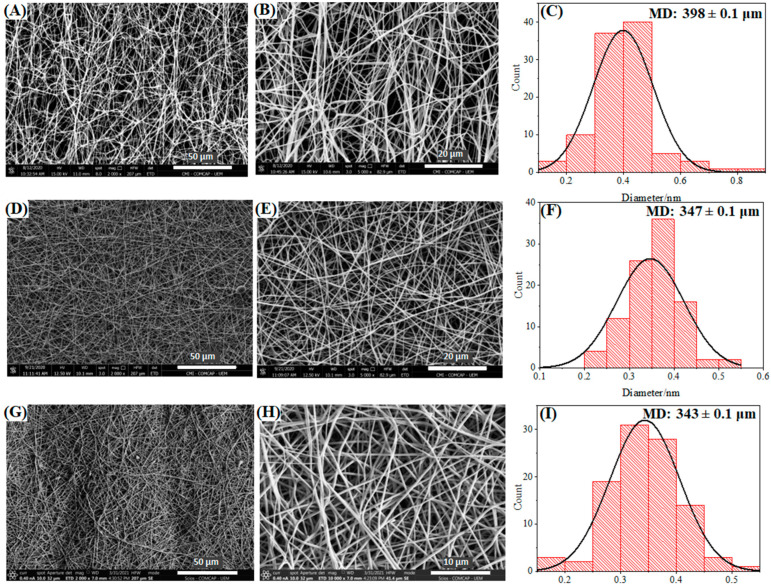
SEM images and diameter distributions of nanofibers: (**A**–**C**) PVA/AgNPs, (**D**–**F**) PVA/CHT/AgNPs 70:30, and (**G**–**I**) PCL [PVA/CHT/AgNPs] coaxial.

**Figure 5 pharmaceutics-16-00268-f005:**
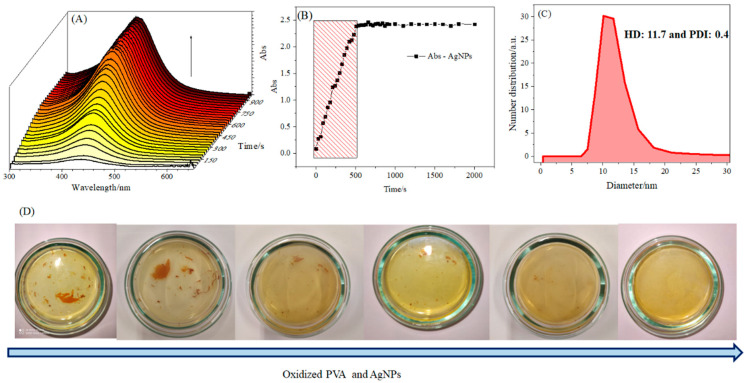
The disintegration of PVA/AgNPs nanofibers in ultrapure water: (**A**) electronic absorption spectra as a function of time, (**B**) kinetic representation, (**C**) DLS of released AgNPs, and (**D**) nanofibers in solution over time.

**Figure 6 pharmaceutics-16-00268-f006:**
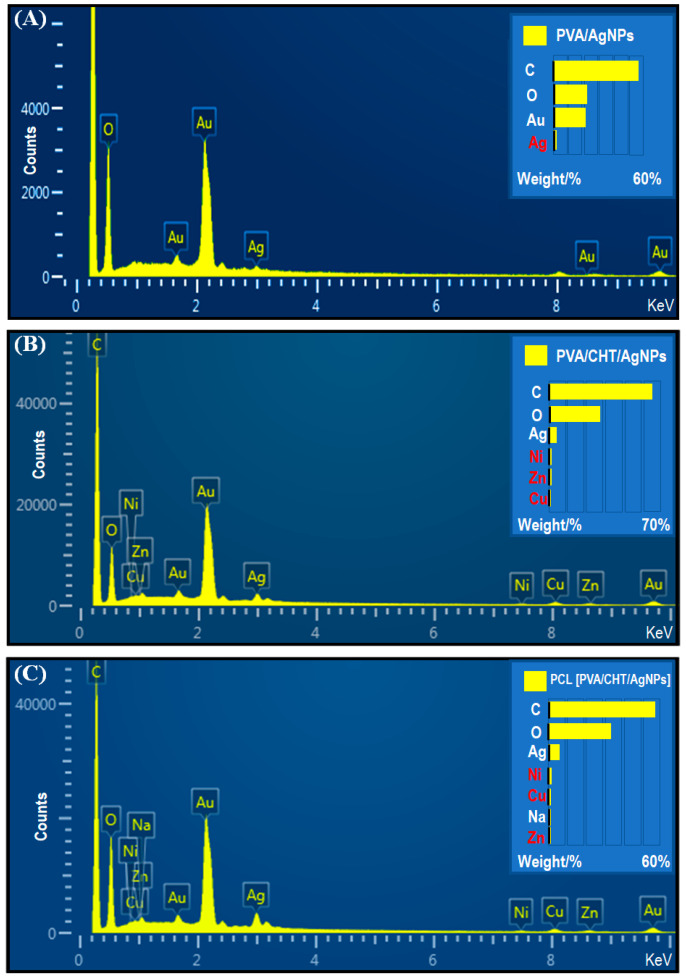
EDS spectra of AgNP-incorporated nanofibers: (**A**) PVA/AgNPs; (**B**) PVA/CHT/AgNPs; and (**C**) PCL [PVA/CHT/AgNPs].

**Figure 7 pharmaceutics-16-00268-f007:**
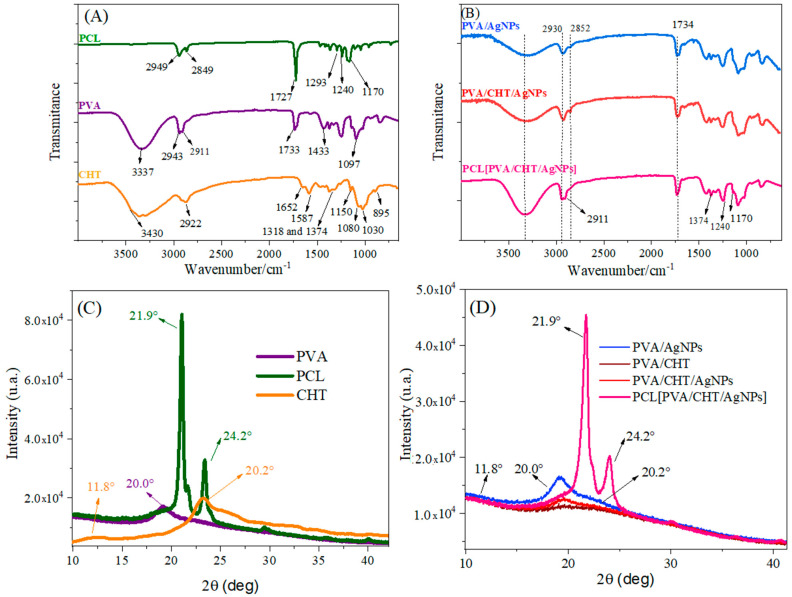
(**A**,**B**) Fourier transform infrared spectroscopy (FTIR-ATR) spectra of nanofibers; (**C**,**D**) DRX spectra of nanofibers and their precursors.

**Figure 8 pharmaceutics-16-00268-f008:**
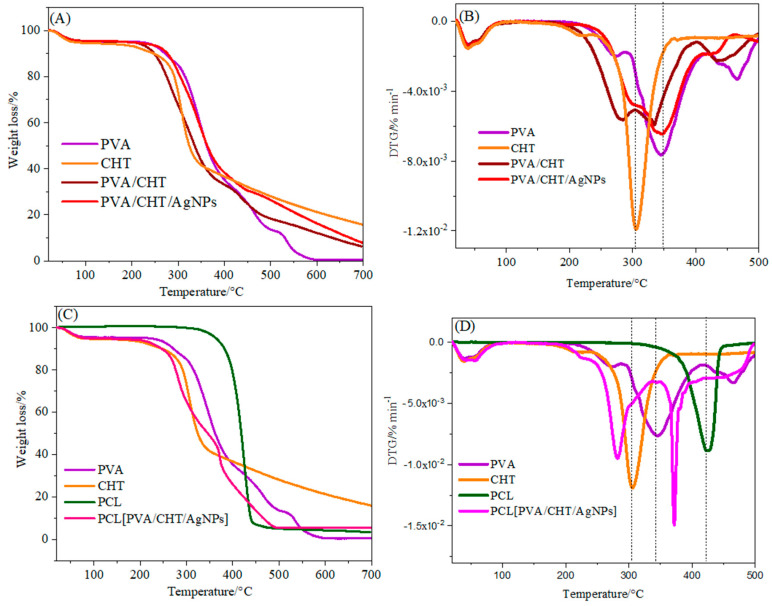
(**A**,**C**) Thermogravimetric analysis, TGA, and (**B**,**D**) differential thermogravimetric analyses, DTG, of nanofibers and their precursors.

**Table 1 pharmaceutics-16-00268-t001:** Preparation of solutions for electrospinning.

Solution	Material	%(w/V)	Solvent	Conditions
1	PCL	10.0	DMF/DCM (1:1)	1 h at 25 °C
2	CHT	10.0	H_2_O and 2% CH_3_COOH	3 h at 60 °C
3	PVA	12.0	H_2_O	3 h at 80 °C *
4	PVA-AgNO_3_	12.0; 0.25	H_2_O	3 h at 80 °C *

* Reflux.

**Table 2 pharmaceutics-16-00268-t002:** Mechanical properties of nanofibers: Young’s modulus, tensile strength, and elongation at break.

Sample	Young’s Modulus (MPa)	Tensile Strength (10^−3^ MPa)	Elongation at Break (%)
PVA	0.164 ± 0.008	10.8 ± 1.0	21.9 ± 6.9
PVA AgNPs	0.084 ± 0.001	9.3 ± 0.10	113.6 ± 15.6
PVA/CHT	0.020 ± 0.004	1.25 ± 0.10	6.2 ± 2.0
PVA/CHT/AgNPs	0.030 ± 0.003	9.50 ± 0.05	15.5 ± 2.4
PCL[PVA/CHT]	0.013 ± 0.005	1.45 ± 0.05	12.1 ± 0.3
PCL[PVA/CHT/AgNPs]	0.049 ± 0.002	3.70 ± 0.70	16.1 ± 1.5

**Table 3 pharmaceutics-16-00268-t003:** Assays with coronavirus (MHV-3) at different contact times with the tested samples. Results are expressed in log reduction (Log ± SD).

Time (h)	PVA	PVA/CHT	PCL[PVA/CHT]	AgNPs	PVA/AgNPs	PVA/CHT/AgNps	PCL[PVA/CHT/AgNPs]
	Log ± SD	Log ± SD	Log ± SD	Log ± SD	Log ± SD	Log ± SD	Log ± SD
0.5	0.8 ± 0.5	1.5 ± 0.5	1.50 ± 0.5	5.3 ± 0.5	5.5 ± 0.5	5.3 ± 0.4	5.0 ± 0.7
1	1.0 ± 0.0	1.8 ± 0.4	1.50 ± 0.5	5.3 ± 0.5	5.5 ± 0.5	5.3 ± 0.4	5.3 ± 0.4
8	1.0 ± 0.4	2.0 ± 0.7	1.75 ± 0.4	5.8 ± 0.5	5.3 ± 0.4	5.5 ± 0.5	5.5 ± 0.5
24	0.8 ± 0.4	2.0 ± 0.0	2.00 ± 0.0	5.3 ± 0.5	5.5 ± 0.5	5.5 ± 0.5	5.5 ± 0.5
48	1.0 ± 0.0	2.0 ± 0.0	2.00 ± 0.0	6.5 ± 0.6	5.5 ± 0.5	5.5 ± 0.5	5.5 ± 0.5
64	1.0 ± 0.0	2.0 ± 0.0	2.00 ± 0.0	6.8 ± 0.5	5.5 ± 0.5	5.5 ± 0.5	5.5 ± 0.5

## Data Availability

Data will be available under request.

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
