# Peer review of "Silver Nanoparticles In Situ Synthesized and Incorporated in Uniaxial and Core–Shell Electrospun Nanofibers to Inhibit Coronavirus"

_pharmaceutics, 2024, doi:10.3390/pharmaceutics16020268_

Round 1

Reviewer 1 Report

Comments and Suggestions for Authors

The present work reported using the electrospinning method to synthesize uniaxial and coaxial nanofiber matrices (NMs) integrated with Ag nanoparticles (NPs). Ag NPs with an average size of 11.6 nm were formed and stabilized inside the nanofibers through an in-situ reduction with PVA – an essential component of nanofibers as the reducing agent. The developed NMs exhibited potent virucidal activity against the mouse hepatitis virus (MHV-3) genus Betacoronavirus strain with log’s reduction ≥5, which is accompanied by non-cytotoxicity. This work is subjected to critical issues that must be carefully revised before proceeding with the next steps.

1.     The description of the virucidal assay is too long. The authors should provide a concise description focusing mainly on the testing process for developed nanofiber matrices. The procedures for the preparation of culture as well as the testing process for referent samples such as AgNPs should be put in Supporting Information.

2.     The scale bar of SEM images in Figure 2 is too small. A modification should be made. The unit of the x-axis of charts in Figures 2C, F and I is incorrect. It must be μm (not nm). The same issues are observed in Figure 5.

3.     Photos of prepared samples in Figure 3 should be put in Supporting Information.

4.     The authors measured the size of Ag NPs (11.7 nm) in PVA/AgNPs sample and concluded that it is also the size of NPs in other samples. This conclusion must be carefully verified. Chitosan is also a reducing agent and a surfactant, in addition to PVA. Its presence can have impact on the formation of AgNPs, including their size (Appl Biochem Microbiol 58, 97–104 (2022)). For this reason, the size of AgNPs in all samples containing them should be investigated. TEM images of released AgNPs should be provided to confirm their size. The role of chitosan in the formation of AgNPs should also be mentioned in the manuscript.

5.     According to the authors, the release of AgNPs plays an essential role in virucidal activity. For this reason, the release of AgNPs from all prepared samples when brought in contact with the biological culture should be investigated as the function of time. The amount of released AgNPs should also be quantified.    

6.     The EDS analysis indicates the significant presence (not trace as the authors said) of other metals such as Ni, Cu, and Zn (Figure 7B and C) in two samples. Why do the two samples contain these impurities?  

7.     Table 3 contains the results of a previous study. It should not be put in the manuscript. Tables and Figures should only include results obtained from this work.

8.     The virucidal activity of prepared NM should be investigated after several testing cycles. This is an essential parameter for evaluating the biological durability of prepared materials. Materials with virucidal activity, which is conserved after many uses, would have a high potential for the practical fabrication of protective equipment.

9.   Too many references (96) were cited in this manuscript. Refs are unnecessary in many places, for example, “PVA/CHT/AgNPs nanofibers, Figure 7B, … trace amounts, probably complexed with CHT [65]”. The number of references must be reduced (50 refs are sufficient for a research article).

Comments on the Quality of English Language

The manuscript contains several grammatical errors that need to be corrected.

Author Response

Reviewer#1:

The present work reported using the electrospinning method to synthesize uniaxial and coaxial nanofiber matrices (NMs) integrated with Ag nanoparticles (NPs) Ag NPs with an average size of 11.6 nm were formed and stabilized inside the nanofibers through an in-situ reduction with PVA- an essential component of nanofibers as the reducing agent. The developed NMs exhibited potent virucidal activity against the mouse hepatitis virus (MHV-3) genus Betacoronavirus strain with log's reduction 25, which is accompanied by non-cytotoxicity. This work is subjected to critical issues that must be carefully revised before proceeding with the next steps.

Response:

Dear Reviewer, thank you for your insightful evaluation of our manuscript. Your intuitive observations and valuable suggestions have played a pivotal role in enhancing the quality and clarity of our study. Your contribution is indispensable to refining this work, and we are committed to addressing each point with utmost care and thoroughness.

  1. The description of the virucidal assay is too long. The authors should provide a concise description focusing mainly on the testing process for developed nanofiber matrices. The procedures for the preparation of culture as well as the testing process for referent samples such as AgNPs should be put in Supporting Information.

Response: Thank you once again for your invaluable contribution to the improvement of our work. We have carefully considered your suggestion to streamline the description of the virucidal assay, focusing mainly on the testing process for the developed nanofiber matrices. Additionally, as per your recommendation, we have moved the procedures for the preparation of culture and the testing process for referent samples, such as AgNPs, to the Supporting Information section.

  1. The scale bar of SEM images in Figure 2 is too small. A modification should be made. The unit of the x-axis of charts in Figures 2C, F and I is incorrect. It must be um (not nm). The same issues are observed in Figure 5.

Response: We appreciate your insightful review and constructive feedback on our manuscript. In response to your suggestions, we have made the following modifications:

  • The scale bar of SEM images in Figures 2 and 5 has been appropriately adjusted to ensure clarity.
  • The unit of the x-axis in charts has been corrected to "μm" instead of "nm."

  1. Photos of prepared samples in Figure 3 should be put in Supporting Information.

Response: Thank you for your helpful feedback on our manuscript. We've made the suggested changes, moving the photos from Figure 3 to the Supporting Information, as you recommended (New Figure S3).

  1. The authors measured the size of Ag NPs (11.7 nm) in PVA/AgNPs sample and concluded that it is also the size of NPs in other samples. This conclusion must be carefully verified. Chitosan is also a reducing agent and a surfactant, in addition to PVA. Its presence can have impact on the formation of AgN*P_{S} including their size (Appl Biochem Microbiol 58, 97-104 (2022)). For this reason, the size of AgNPs in all samples containing them should be investigated. TEM images of released AgNPs should be provided to confirm their size. The role of chitosan in the formation of AgNPs should also be mentioned in the manuscript.

Response: Thank you for your insightful comments.

Indeed, the impact of chitosan on the formation of AgNPs was one of the focal points carefully considered during the development and composition of the manuscript. This contribution was included and highlighted in the manuscript (See page 18, lines 16 to 19). However, it is worth noting that in the present study, a low concentration of chitosan (30%, 0.35 mmol L-1) was employed in comparison to PVA (70%, 0.80 mmol L-1). Additionally, the prior contact of PVA with silver nitrate facilitates the complexation of Ag+ with PVA, to obtain [Ag+PVA] as discussed in the manuscript. Furthermore, the PVA/CHT mixture was conducted at room temperature. Nevertheless, the literature reports that chitosan can function as a reducing agent in the presence of heating at 95 °C for 12 hours [1] to obtain AgNPs with sizes between 6-8 nm. Another study demonstrated the necessity of treating a solution of chitosan and silver nitrate with ultrasound for 1 hour at a temperature of 60–90°C to obtain nanoparticles. Moreover, an increase in AgNPs size was observed with a rise in the reaction temperature [2].

Therefore, based on the in vitro results, it is unlikely that AgNPs underwent drastic alterations due to the influence of chitosan. Changing the size of nanoparticles would certainly imply significant changes in in vitro results, a fact that was not observed.

[1]   D. Wei, W. Sun, W. Qian, Y. Ye, X. Ma, The synthesis of chitosan-based silver nanoparticles and their antibacterial activity, Carbohydr. Res. 344 (2009) 2375–2382. https://doi.org/10.1016/j.carres.2009.09.001.

[2]   G.M. Raghavendra, J. Jung, D. Kim, K. Varaprasad, J. Seo, Identification of silver cubic structures during ultrasonication of chitosan AgNO3 solution, Carbohydr. Polym. 152 (2016) 558–565. https://doi.org/10.1016/j.carbpol.2016.07.045.

  1. According to the authors, the release of AgNPs plays an essential role in virucidal activity. For this reason, the release of AgNPs from all prepared samples when brought in contact with the biological culture should be investigated as the function of time. The amount of released AgNPs should also be quantified.

Response: Thank you for your thoughtful review and constructive feedback on our manuscript. We appreciate the time and effort you invested in evaluating our work.

As demonstrated in Figure 4, only the AgNPs obtained in the hydrophilic PVA nanofiber showed disintegration. However, the other nanofibers containing chitosan and PCL did not exhibit disintegration. Indeed, release assays were conducted, but the release of AgNPs was not observed. Nevertheless, in vitro assays revealed the efficient virucidal activity of the prepared nanofibers, even in cases where the release of AgNPs was not evident. This outcome holds potential positivity when considering the intended application. In this instance, the material is intended for the production of collective and individual protective equipment, aiming to achieve surface protection.

Therefore, the AgNPs must be distributed throughout the ultrafine and porous material, such as the nanofibers, allowing their contact with fluids contaminated with viruses (for example, from sneezes). Thus, as previously discussed in the literature, the dissolution of AgNPs and the release of Ag+ ions into the relevant medium represent a plausible mechanism for antimicrobial activity [3]. In terms of surface applications, the presence of immobilized nanoparticles capable of serving as reservoirs for silver ions is noteworthy. In this manner, immobilized AgNPs continuously provide a sufficiently high concentration of silver antimicrobial species in their vicinity, maintaining activity for several days, as desired in collective and individual protective materials.

This effect can be highly advantageous in real-world application conditions, where fluid circulation on-site leads to the release of active species. Thus, an immobilized nanoparticle near a microorganism can release several tens of thousands of silver atoms in its vicinity, creating a locally elevated concentration of antimicrobial agents (Trojan horse effect)  [4].

To enhance the clarity of the manuscript, this discussion has been incorporated into the text (see page 18, lines 20-31).

[3]          B. Le Ouay, F. Stellacci, Antibacterial activity of silver nanoparticles: A surface science insight, Nano Today. 10 (2015) 339–354. https://doi.org/10.1016/j.nantod.2015.04.002.

[4]          E.J. Park, J. Yi, Y. Kim, K. Choi, K. Park, Silver nanoparticles induce cytotoxicity by a Trojan-horse type mechanism, Toxicol. Vitr. 24 (2010) 872–878. https://doi.org/10.1016/j.tiv.2009.12.001.

  1. The EDS analysis indicates the significant presence (not trace as the authors said) of other metals such as Ni, Cu, and Zn (Figure 7B and C) in two samples. Why do the two samples contain these impurities?

Response: Thank you for raising this issue. The identified impurities, specifically Ni, Cu, and Zn, are indeed detected in two samples owing to the inclusion of chitosan.

The occurrence of Ni, Cu, and Zn can be traced back to the chitosan utilized in the synthesis process, given its inherent capability to readily complex with these ions [5]. We acknowledge your thoroughness in pinpointing this inconsistency, and we have revised the manuscript to precisely elucidate the origin of these impurities.

To enhance the clarity of the manuscript, this discussion has been incorporated into the text (see page 19, lines 8-9).

[5]          J.R.B. Gomes, M. Jorge, P. Gomes, Interaction of chitosan and chitin with Ni, Cu and Zn ions: A computational study, J. Chem. Thermodyn. 73 (2014) 121–129. https://doi.org/10.1016/j.jct.2013.11.016.

  1. Table 3 contains the results of a previous study. It should not be put in the manuscript Tables and Figures should only include results obtained from this work

Response: Thank you for your insightful feedback. We have carefully considered your suggestion and made the necessary adjustments. The data originally presented in Table 3, which pertains to Laboratory, M. Log, and Percent Reductions in Microbiology and Antimicrobial Testing, functions as a benchmark for categorizing the applied materials based on their virucidal properties. The Table has been relocated to the Supplementary Material (Table S1). This modification ensures that our manuscript Tables and Figures now exclusively include results obtained from the current study.

8 The virucidal activity of prepared NM should be investigated after several testing cycles. This is an essential parameter for evaluating the biological durability of prepared materials. Materials with virucidal activity, which is conserved after many uses, would have a high potential for the practical fabrication of protective equipment.

Response:  We sincerely appreciate your suggestion regarding the investigation of the virucidal activity of the prepared nanofiber matrices after several testing cycles. Your emphasis on the biological durability of the materials is indeed a critical parameter for practical applications. It is anticipated that the immobilization of AgNPs on the nanofiber will extend the action time of the materials. However, we would like to note that such experiments may involve complexities and resource requirements that go beyond the scope of the current study. While conducting tests over multiple cycles is not readily achievable within the constraints of this particular work, we fully recognize the validity of your suggestion. Investigating the performance of the nanofiber matrices over extended use is an avenue we intend to explore in subsequent studies.

  1. Too many references (96) were cited in this manuscript. Refs are unnecessary in many places, for example, "PVA/CHT/AgNPs nanofibers, Figure 78, trace amounts, probably complexed with CHT 1657. The number of references must be reduced (50 refs are sufficient for a research article).

Response: Thank you for your feedback. We appreciate your insight regarding the number of references cited, and we have taken your suggestion to heart.

In response, we have diligently revised the manuscript, reducing the number of references without compromising the quality and reliability of the information presented. We believe this adjustment enhances the clarity and conciseness of our work while maintaining the necessary scholarly support for our findings.

  1. The manuscript contains several grammatical errors that need to be corrected.

Response: Thank you for your commentary. We have carefully reviewed the manuscript and addressed the grammatical errors. The revised version has been thoroughly examined by a native English speaker to ensure the accuracy and fluency of the language. We appreciate your valuable input and hope that the revisions meet your expectations.

Reviewer 2 Report

Comments and Suggestions for Authors

It is a nice and original work. It would be nice and helpful if the authors will manage to present some concrete examples of possible applications of these research in introduction and in conclusions too. They should correlate for eg. the analysis of mechanical properties with the possible applications.

Author Response

Reviewer 2:

It is a nice and original work. It would be nice and helpful if the authors will manage to present some concrete examples of possible applications of these research in introduction and in conclusions too. They should correlate for eg. the analysis of mechanical properties with the possible applications.

Response: We would like to express our thanks for your insightful comments on our manuscript, and we appreciate the positive feedback on the overall quality of the work.

Regarding your suggestion to present concrete examples of possible applications in the introduction and conclusion, we would like to inform you that we have, carefully considered your input and have made the necessary revisions to address this point (see page 5, lines 8-15, page 27, lines 29-34, and page 28, line 1). Thank you once again for your time and consideration.

Round 2

Reviewer 1 Report

Comments and Suggestions for Authors

The authors have made a thorough revision that has cleared my concerns. I recommend publication of this work in its present form.